# Changes of In Situ Prokaryotic and Eukaryotic Communities in the Upper Sanya River to the Sea over a Nine-Hour Period

**DOI:** 10.3390/microorganisms11020536

**Published:** 2023-02-20

**Authors:** Shijie Bai, Jian Zhang, Xiaoxue Qi, Juntao Zeng, Shijun Wu, Xiaotong Peng

**Affiliations:** 1Institute of Deep-Sea Science and Engineering, Chinese Academy of Sciences, Sanya 572000, China; 2The State Key Laboratory of Fluid Power & Mechatronic Systems, Zhejiang University, Hangzhou 310027, China; 3University of Chinese Academy of Sciences, Beijing 100049, China

**Keywords:** microbiome, abundant communities, rare communities, in situ filtration system, stochastic processes

## Abstract

The transition areas of riverine, estuarine, and marine environments are particularly valuable for the research of microbial ecology, biogeochemical processes, and other physical–chemical studies. Although a large number of microbial-related studies have been conducted within such systems, the vast majority of sampling have been conducted over a large span of time and distance, which may lead to separate batches of samples receiving interference from different factors, thus increasing or decreasing the variability between samples to some extent. In this study, a new in situ filtration system was used to collect membrane samples from six different sampling sites along the Sanya River, from upstream freshwater to the sea, over a nine-hour period. We used high-throughput sequencing of 16S and 18S rRNA genes to analyze the diversity and composition of prokaryotic and eukaryotic communities. The results showed that the structures of these communities varied according to the different sampling sites. The α-diversity of the prokaryotic and eukaryotic communities both decreased gradually along the downstream course. The structural composition of prokaryotic and eukaryotic communities changed continuously with the direction of river flow; for example, the relative abundances of *Rhodobacteraceae* and *Flavobacteriaceae* increased with distance downstream, while *Sporichthyaceae* and *Comamonadaceae* decreased. Some prokaryotic taxa, such as *Phycisphaeraceae* and *Chromobacteriaceae*, were present nearly exclusively in pure freshwater environments, while some additional prokaryotic taxa, including the *SAR86 clade*, *Clade I*, *AEGEAN-169 marine group*, and *Actinomarinaceae*, were barely present in pure freshwater environments. The eukaryotic communities were mainly composed of the *Chlorellales X*, *Chlamydomonadales X*, *Sphaeropleales X*, *Trebouxiophyceae XX*, *Annelida XX*, and *Heteroconchia*. The prokaryotic and eukaryotic communities were split into abundant, common, and rare communities for NCM analysis, respectively, and the results showed that assembly of the rare community assembly was more impacted by stochastic processes and less restricted by species dispersal than that of abundant and common microbial communities for both prokaryotes and eukaryotes. Overall, this study provides a valuable reference and new perspectives on microbial ecology during the transition from freshwater rivers to estuaries and the sea.

## 1. Introduction

Microbial communities, including bacteria, archaea, eukaryotes, and viruses, are extremely diverse at the taxonomic and metabolic levels, and play a key role in global biogeochemical processes [1,2,3,4,5]. Microorganisms are major players and drivers of nutrient cycling in rivers, and the role of rivers and streams in global biogeochemical cycling is now recognized as involving the active transformation of organic compounds rather than their passive transport to the ocean [6,7,8,9]. Some lotic ecosystems, such as rivers and streams, are distinctive, since they represent spatial and temporal continuity during their transition from headwaters to estuaries. Recent studies have uncovered the diversity, dynamics and spatial heterogeneity of microbial communities in rivers [10,11]. Moreover, as a previous study on the Yangtze River pointed out, the main factor affecting the bacterial communities was the seasonal changes in temperature in rivers [12]. Therefore, the temperature in different seasons had a significant effect on the alpha diversity of river microorganisms, and similar conclusions were reached in a later study on the Saint Charles River [11]. Observations in riverine microbial systems showed a variety of phenomena in which alpha diversity could be either positively or negatively related along the river from upstream to downstream [10,13].

The relationship between the distance from upstream to downstream and the diversity of organisms in the river is not always clear and may depend on the river system and the types of organisms being examined. A strong relationship of increasing fish biological diversity with decreasing distance from the river outlet has been previously documented [14,15]. The diversity of aquatic insect communities in rivers is correlated with catchment area, and has shown similar patterns to fish in rivers [16]. Nevertheless, prokaryotic diversity tends to decrease gradually along the river from upstream to downstream [17]. However, based on aquatic environmental DNA (eDNA) technology [18], the diversity of riverine bacteria increased with distance downstream [15] while invertebrate diversity in the river remained spatially constant [15]. However, there is uncertainty in the detection distance of eDNA delivered downstream through the river [19], especially in fast-flowing rivers, which can transport eDNA downstream over large distances before its degradation and dilution [20]. Therefore, it remains inconclusive as to whether the diversity of prokaryotic and eukaryotic communities is increasing or decreasing along the river from upstream to downstream.

While studies on eukaryotic microbial communities in river water are relatively limited [21,22,23], prokaryotic microbial communities, especially bacterial communities, in rivers are well documented, but most of these studies have focused on sediments [12,24,25]. Nevertheless, the rules associated with more lentic habitats, such as sediments or even static environments, may not be applicable to an ever-changing lotic environment, such as rivers, which represent an under-researched habitat. Furthermore, microorganisms, unlike many macroorganisms, are usually passive dispersers [26], with the direction of dispersal determined by the flow of water. Rivers also have a large number of potential inputs, each of which adds a new microbial community to the mixture. For example, at any point in a river system, most of the water, and, therefore, the microbial communities, are likely to come from upstream. The concept of River Continuum described by Vannote and colleagues in 1980 proposed that biological communities in rivers use nutrients converted by upstream communities to gradually change from the headwaters to the estuary [27]. However, rivers may also be subject to anthropogenic pressures, such as passing through urbanized, industrialized, agricultural or forested areas, with their associated potential effects on the composition and formation processes of microbial communities in the rivers [28]. Not only that, changes in hydrological conditions, such as soil and runoff inputs [29], tropical storms, and atmospheric deposition [30], will also have an impact. Moreover, microbial communities in extreme and simple ecological environments are more susceptible to impacts, and these effects exhibit a clear seasonality [31].

At the river-to-estuary scale, water movement is thought to play a key role in regulating species taxonomy and microbial community assemblages, because the movement of water and microorganisms is directional—from river to sea, most ecosystems are both recipients and sources of microorganisms, and the migrant pools that reach a particular ecosystem will vary depending on their location in the hydrological continuum and some anthropogenic or natural factors, which can lead to strong linkages and the intense exchange of microorganisms between local and exogenous input communities [32,33,34]. Thus, to better study the changes in microbial communities in river to marine waters, especially considering that we do not have a full understanding of the evolutionary relationships between marine and freshwater microbes [35], in situ sampling at different locations from river to the sea over a very short period of time can provide new research perspectives. Another point to note is that any microbial community includes abundant taxa, rare taxa, and other taxa that, together, make up a complete microbial community. Several of these species are highly abundant, other species are moderately abundant, and a large number of species have only a few individuals [36,37]. The characteristics expected of abundant and rare taxa suggest that abundant taxa constitute the vast majority of biomass in the community and play a major role in carbon and nutrient cycling, whereas rare taxa dominate community biodiversity. Rare taxa confer an almost inexhaustible reservoir of microbial genetic material, help to maintain microbial diversity, and provide a mechanism for the diverse microbial assemblages that allow for the persistence of microorganisms in a community, thus contributing to community resilience. Rare taxa also contribute to a significant proportion of auxiliary functions, which suggests that rare taxa may be a source of functional diversity and, thus, contribute to ecosystem stability [36,37,38]. All taxa found in the rare biosphere must have grown in the past, may currently grow in a different location, or experience both types of growth. When environmental conditions become favorable, a rare taxon will grow and become abundant members of the community while the resources it excels at acquiring are consumed, and then the taxon is eliminated by the loss process back to the rare biosphere [39]. The distribution of abundant and rare taxa must be dynamic, with members of the abundant fraction decreasing toward the rare biosphere, and vice versa, as environmental conditions change frequently, such as through seasonal changes or other episodic factors, which include, for instance, heavy rains, storms, or typhoons [36,40]. Therefore, if sufficient samples can be collected within a very short period of time, it would better reveal more realistic information about the microbial community in a particular environment.

To date, the vast majority of studies of microbial communities in rivers or estuaries have been based on samples collected at different temporal and spatial scales, including those which may have been collected over long periods of time and have thus been subject to many disturbing factors. In contrast, studies of in situ prokaryotic and eukaryotic communities in the river-to-sea continuum that are little disturbed by external factors within a short time period are scarce. To address this research gap, we collected 12 samples from along the upper Sanya River to the sea using an in situ filtration system to characterize changes in prokaryotic and eukaryotic communities over a nine-hour period. The Illumina Hiseq platform was used to perform high-throughput DNA sequencing of partial 16S and 18S rRNA genes, thus describing the composition and structure of the in situ prokaryotic and eukaryotic communities.

## 2. Materials and Methods

### 2.1. Sample Collection

The Sanya River, located in the southern part of Sanya city, passes through the city from north to south and discharges into the sea at Sanya Port, with a total length of 28.8 km and a catchment area of 337.02 square kilometers. We previously set up five sampling sites along the Sanya River, from the upstream section to the estuary, and collected one 3 um and one 0.22 µm polycarbonate (PC) membrane (Φ 147 mm, Millipore, Burlington, MA, USA) at each sampling point using an in situ filtration system (Figure 1). At the same time, we also set up a sampling site at the seashore ~6 km from the estuary, and collected one sample each of 3 µm and 0.22 µm filter membranes on 22 January 2022. In total, there were six sampling sites with 12 filter samples (Figure 2). The filter membranes containing particle-associated and free-living microorganisms were folded into separate sterile 15 mL centrifuge tubes, packed in dry ice for shipping to the laboratory, and then stored at −80 °C until analysis.

### 2.2. DNA Extraction and Sequencing

The filter membranes were cut into small pieces with sterile scissors prior to nucleic acid extraction. DNA was extracted from different filter membrane samples (three extraction blank control samples were used) with DNeasy PowerSoil Pro Kit (QIAGEN, Germantown, MD, USA), according to the manufacturer’s instructions. The extracted DNA was quantified with a Qubit fluorometer (Invitrogen Inc. Manufacturer: Life Technologies Holdings Pte Ltd., Singapore) and used for amplification of the V4 region of the 16S rRNA gene with the primer pair 515f Modified (5′GTGYCAGCMGCCGCGGTAA-3′) and 806r Modified (5′GGACTACNVGGGTWTCTAAT-3′) [41], as well as of the V4 region of the 18S rRNA gene with the primer pair 3NDF (5′-GGCAAGTCTGGTGCCAG-3′) and V4-euk-R2R (5′-ACGGTATCTRATCRTCTTCG-3′) [42]. The 16S rRNA gene PCR cycling conditions were as follows: denaturation at 95 °C for 3 min, followed by 27 cycles at 95 °C for 30 s, 55 °C for 30 s, 72 °C for 45 s, and a final extension at 72 °C for 10 min. Triplicate PCR amplicons were combined after purification using a TaKaRa purification kit (TaKaRa, Kusatsu, Shiga, Japan). 18S rRNA gene PCR cycling conditions were consistent with those of the 16S rRNA gene, except that 32 cycles were performed. The PCR products were prepared for library construction with the TruSeq DNA sample preparation kit (Illumina, San Diego, CA, USA) according to the manufacturer’s instructions. The libraries were sequenced at MajorBio Co. Ltd. (Shanghai, China) using the HiSeq platform (Illumina) with a paired-end 300 bp sequence read run. The raw sequencing reads of all samples were deposited in the NCBI database (http://www.ncbi.nlm.nih.gov/ (accessed on 27 December 2022)) under BioProject accession number: PRJNA915948 for prokaryotic and eukaryotic community datasets.

### 2.3. Microbial Community Analysis

After sequences were assigned to their respective samples, barcodes as well as forward and reverse primers (one mismatch each was allowed) were removed to obtain clean data. The FLASH program (v. 1.2.8) [43] was used to obtain paired-end sequences of sufficient length, with at least a 30 bp overlap combined into full-length sequences, and the average fragment length was 253 bp for prokaryotes and 435 bp for eukaryotes. High-quality sequences, without ambiguous bases (Ns) were retained using the Btrim program (v. 0.2.0), and sequences between 245 bp and 260 bp in length for prokaryotes and from 415 bp to 435 bp for eukaryotes were used for the following analyses [44]. UNOISE3 was applied to generate amplicon sequence variants (ASVs) with default settings [45]. A representative sequence from each ASV of 16S rRNA gene sequencing was selected for taxonomic annotation via comparison with the SILVA 132 database [46], which includes bacterial, archaeal, and eukaryotic sequences, while the representative sequence from each ASV of 18S rRNA gene sequencing was annotated by comparison with the PR2 database (v. 4.14.0) [47]. The generated ASV table was used in the following analyses. Following guidelines from previous studies, ASVs with relative abundances > 1% in one sample were classified as ‘abundant’, ASVs with relative abundances < 0.1% across all samples were defined as ‘rare’, and the remaining ASVs with intermediate abundance were defined as ‘common’ [48,49]. To take into account the different sequencing depths, ASVs were randomly resampled to normalize the reads for each sample. The diversity of prokaryotic and eukaryotic microbial communities from the filter membrane samples of different sampling sites was determined via statistical analysis of the α-diversity indices (Shannon, Inverse Simpson, Chao1) [50] and observed richness. R language version 3.4.3 [51] and Mothur program [52] were used to calculate these α-diversity indices. ß-diversity-based statistical tools and non-metric multidimensional scaling (NMDS), were used to test the differences within the prokaryotic and eukaryotic microbial community structures.

### 2.4. Prokaryotic and Eukaryotic Microbial Community Assembly Predicted by Neutral Community Model

Several previous studies have shown that stochastic events (births, deaths and immigration) play a key role in shaping the structure of different communities, including those of macroorganisms, prokaryotes, and eukaryotes. Neutral community models (NCMs) have been shown to successfully reproduce the observed abundance distributions of species communities and can therefore be used to predict prokaryotic and eukaryotic community assembly mechanisms [49,53,54,55]. In the neutral model, R^2^ represents the overall goodness-of-fit of the model, with higher R^2^ values indicating that it is closer to the neutral model, i.e., the construction of the community is more influenced by stochastic processes and less influenced by deterministic processes, while N describes the metacommunity size, which, in this paper, is the total abundance of all OTUs in each sample, and m quantifies the migration rate at the community level, which is uniform for each community member (independent of species). A smaller value of m indicates a more restricted species dispersal throughout the community, and, conversely, a higher value of m indicates a less restricted species dispersal. Nm is the product of metacommunity size (N) and migration rate (m), which quantifies the estimate of dispersal between communities and determines the correlation between the frequency of occurrence and relative regional abundance [54,55]. The 95% confidence intervals for all fitted statistics were determined by using the Hmisc package with 1000 bootstrap replicates in R [56].

## 3. Results and Discussion

### 3.1. Sequencing Statistics and Microbial Diversity

A total of 541,658 prokaryotic sequences and 590,031 eukaryotic sequences were obtained from the 12 in situ filtration samples, with separate filter samples having retained the particle-associated and free-living microorganisms, respectively, for each site. An average of 45,138 ± 6465 prokaryotic sequences and 49,169 ± 12,142 eukaryotic sequences were obtained from each sample. In order to obtain a more accurate result of α-diversity, we rarefied to 39,024 prokaryotic sequences and 29,690 eukaryotic sequences per sample for analysis of the prokaryotic and eukaryotic diversity, composition, and structure. The α-diversities of the original, abundant, rare, and common microbial communities from the 12 in situ filtration samples were calculated. The Shannon, Inverse Simpson, and Chao1 indices and observed richness all indicated that the α-diversity of the prokaryotic and eukaryotic microbial communities decreased gradually along the Sanya River from upstream to downstream (Table 1 and Table 2).

### 3.2. Structure and Composition of the Microbial Communities

NMDS analysis of prokaryotic and eukaryotic communities clearly showed six principal groups that corresponded to the sampling sites of Sanya 1, Sanya 2, Sanya 3, Sanya 4, Sanya 5, and Sanya 6 (Figure 3 and Figure 4). These results suggest that microbial community structure differed between sampling sites, with the prokaryotic community structure mainly influenced by rare microbial taxa, whereas the eukaryotic community structure was influenced by a combination of abundant, rare and common microbial taxa.

The relative abundance of prokaryotes was evident at the phylum, family, and genus levels, with a similarity of 97% for ASV classification, and this provided detailed information on the composition of the microbial communities (Figure 5, Figure 6 and Figure 7). The dominant prokaryotic phyla were Actinobacteriota, Proteobacteria (Gamma), Proteobacteria (Alpha), and Bacteroidota. The phyla Actinobacteriota and Proteobacteria (Gamma) decreased from upstream to downstream, while those of the phyla Proteobacteria (Alpha) and Bacteroidota displayed the opposite trend. At the family level, *Rhodobacteraceae* and *Flavobacteriaceae*, which are affiliated with the Proteobacteria (Alpha) and Bacteroidota, respectively, increased in the downstream direction. The relative abundance of *Sporichthyaceae* and *Comamonadaceae*, which are affiliated with the Actinobacteriota and Proteobacteria (Gamma), respectively, were opposite to the flow direction of the Sanya River. Other prokaryotic taxa, such as *SAR86 clade*, *Clade I*, *AEGEAN-169 marine group*, and *Actinomarinaceae*, were barely present in pure freshwater environments. However, in this study, *Phycisphaeraceae* and *Chromobacteriaceae*, were present almost exclusively in pure freshwater environments. Furthermore, at the level of the genus, a large number of prokaryotic ASVs were labeled as unclassified taxa. The relative abundance of the *hgcI clade*, which are affiliated with the *Sporichthyaceae* (Actinobacteriota), decreased in the downstream direction. The genera with high relative abundances also included *Synechococcus CC9902*, *RS62 marine group*, *Ponticaulis*, *Polynucleobacter*, *Oleibacter*, *NS5 marine group*, *NS3a marine group*, *ML602J-51*, *Hydrogenophaga*, *Cyanobium PCC-6307*, Clade Ib, Clade Ia, CL500-3, *Candidatus Aquiluna*, *Candidatus Actinomarina*, *Alteromonas*, and *Acinetobacter*.

The relative abundance of eukaryotes at the phylum, family, and genus levels, with 97% similarity in ASV classification, was also analyzed. Dinoflagellata, Metazoa, and Chlorophyta were the most dominant groups, accounting for 40% to 91% of the eukaryotic community composition. Meanwhile, Ochrophyta, Cryptophyta, Perkinsea, Fungi, and Apicomplexa were also the dominant eukaryotic lineages in the eukaryotic microbial communities (Figure 8). At the family level, *Chlorellales X*, *Chlamydomonadales X*, *Sphaeropleales X*, and *Trebouxiophyceae XX*, all of which are affiliated with the Chlorophyta, and the *Annelida XX* and *Heteroconchia*, which are affiliated with the Metazoa, together constituted the main taxa of the eukaryotic communities (Figure 9). Furthermore, at the genus level, the eukaryotic community was composed mainly of unclassified and “others”, accounting for 25% to 97% of the relative abundance. Apart from this fraction, *Lysidice*, *Choricystis*, *Ostreococcus*, *Mytilopsis*, *Spermatozopsis*, *Perkinsida XXX*, *Chlamydomonas*, and *Scenedesmus* were the major taxa of the eukaryotic microbial communities in the samples (Figure 10). In addition, we did not find a pattern of variation within the composition of the eukaryotic community along the Sanya River.

### 3.3. The Shared and Unique Prokaryotic and Eukaryotic Lineages of Samples Collected from Different Sampling Sites

To better understand the shared and unique prokaryotic and eukaryotic communities of samples collected from different sites, we explored the differences at the ASV level between samples based on sampling site. Quality control and random resampling of these samples were conducted; the sequence reads were clustered into 504 ASVs at the 97% similarity level for prokaryotic datasets and 713 ASVs for eukaryotic datasets. As shown in Appendix A, the filtration samples collected from the six sites from along the upper Sanya River to the sea contained 237, 16, 72, 10, 30, and 58 unique ASVs, respectively. The prokaryotic lineages of *Phycisphaeraceae*, *Gemmataceae*, and *Cyanobiaceae* were dominant in the unique prokaryotic communities of Sanya 1, which was the furthest upstream site. Subsequently, the dominant prokaryotic taxa in the samples from Sanya 2 were *Chitinophagaceae*, *Spirosomaceae*, *Comamonadaceae*, and *Erysipelotrichaceae*. In the samples of Sanya 3, the dominant prokaryotic lineages changed to *Sulfurimonadaceae*, *Acholeplasmataceae*, *Sulfurovaceae*, and *Arcobacteraceae*. At Sanya 4, the main dominant taxa of the unique prokaryotic communities were *Flavobacteriaceae*, *Rhodobacteraceae*, *PS1 clade*, and *Thiotrichaceae*. In the estuarine region, Sanya 5, the dominant prokaryotic taxa shifted to *Desulfovibrionaceae*, *Thermotaleaceae*, *Fusobacteriaceae*, *Morganellaceae*, and *Oscillatoriaceae*. For the samples of Sanya 6, the unique prokaryotic communities consisted mainly of the dominant prokaryotic taxa *Flavobacteriaceae*, *Puniceicoccaceae*, *Chromatiaceae*, *Microbacteriaceae*, and *Rhodobacteraceae*. Furthermore, 81 ASVs were detected in all samples. Among the prokaryote ASVs shared by all samples, the main lineages were *Alteromonadaceae*, *Microbacteriaceae*, *Rhodobacteraceae*, and *Moraxellaceae* (Appendix A). Meanwhile, for the unique eukaryotic communities, 218, 67, 60, 35, 190 and 118 unique ASVs were found in the sampling sites along the Sanya River (Appendix A). In the samples of Sanya 1, the eukaryotic lineages of *Chlorellales X*, *Radial-centric-basal-Coscinodiscophyceae*, *Trebouxiophyceae XX*, *Colpodellida CHR1*, and *Chlamydomonadales X* were dominant. At Sanya 2, the eukaryotic taxa *Chlamydomonadales X* and *Maxillopoda* were predominant. In the samples of Sanya 3, the dominant eukaryotic taxa were *Chytridiomycetes*, *Gregarinidae*, and *Perkinsida XX*, and, for Sanya 4, the dominant eukaryotic taxa were *Annelida XX* and *Heteroconchia*. In the estuary, Sanya 5, the dominant eukaryotic taxa shifted to *Maxillopoda*. Finally, at Sanya 6, the dominant eukaryotic taxa were *Lecudinidae*, *Eugregarinorida EUG2*, and *Chlorellales X*. In addition, a total of 25 ASVs were shared among all samples, with the dominant eukaryotic taxon being *Chlorellales X* (Appendix A).

### 3.4. The Neutral Model of Ccommunity Assembly Based on Abundant, Common, and Rare Microbial Datasets

The neutral community model (NCM) analyses were performed based on the abundant, common, and rare taxa of prokaryotic and eukaryotic datasets, respectively (Figure 11). Our results showed that the rare microbial communities can be better predicted by NCM than abundant and common microbial communities, and the estimated frequency of occurrence of ASVs had a higher explained rate of community change in relation to the change in their relative abundances. In addition, the m values of rare microbial communities were higher than those of common and abundant microbial communities in both the prokaryotic and eukaryotic communities. These findings indicated that the structure formation of rare microbial communities, whether prokaryotic or eukaryotic, was more inclined to stochastic processes compared to the abundant and common communities, and less restricted by species dispersal.

## 4. Discussion

Microorganisms, both prokaryotic and eukaryotic, are important in ecosystems, and riverine microorganisms are the most fundamental part of riverine biomes and ecosystem structure. They play an important role in the transformation and cycling of nutrients, organic matter formation and decomposition, global material cycling and energy flow, and various other biological processes [57,58,59,60,61,62]. The transformation of material and energy in rivers is carried out under complex water conditions. As such, hydrodynamic conditions are an important factor that distinguishes the study of river microbial ecology from the studies of other aquatic environmental systems. The composition structure, species richness, and diversity index of microbial communities in rivers are the fundamental contents of river microbial ecology research, and the distribution patterns of riverine microorganisms are driven by microbial community assembly mechanisms. Therefore, the potential assembly mechanism of river microbial communities is a key scientific question in the study of river microbial ecology [63].

The study of prokaryotic and eukaryotic communities in river water is relatively limited, and the in situ sampling of different locations along the river course within a short period of time can exclude a large number of disturbance factors, as well as provide new perspectives on the microbial ecology research of rivers and their discharge into estuaries and the sea. In this study, we characterized the changes in prokaryotic and eukaryotic communities over a 9 h period through the collection of 12 samples using a filtration system along the course of the Sanya River followed by high-throughput sequencing. We also split the prokaryotic and eukaryotic datasets into three sub-datasets, abundant, common, and rare, for separate analysis and study. Our results showed that the α-diversity of prokaryotic and eukaryotic communities along the Sanya River gradually decreased from upstream to downstream. Similar results have also been demonstrated in a previous study where 454 pyrosequencing of the benthic biofilms from 114 rivers was performed, and the results showed a decreasing trend in microbial diversity from the headwater source to downstream [17].

In this study, *Phycisphaeraceae* and *Chromobacteriaceae* were present almost exclusively in pure freshwater environments. *Phycisphaeraceae* were also found in stream and lake surface water samples from the Fuglebekken and Revvatnet basins in southern Svalbard [64], and in the sediments in lake Chaohu, which is a large, shallow eutrophic lake in China [65]. The Vogesella, which are affiliated with the *Chromobacteriaceae*, are very common in freshwater rivers or lakes, and are capable of peptidoglycan degradation [66,67]. *SAR86 clade*, *clade I*, *AEGEAN-169 marine group,* and *Actinomarinaceae* were almost absent in pure freshwater environments in this study. *SAR86* is an abundant and widespread heterotroph in the surface ocean that plays a pivotal role in the function of marine ecosystems [68]. *SAR11 clade* of the Proteobacteria (Alpha) is one of the most abundant planktonic bacteria in the marine ecosystem [69]. Meanwhile, *AEGEAN-169 marine group* and *Actinomarinaceae* are also common microbial taxa in the ocean [70,71]. At the taxonomic level of eukaryotes, the *Chlorellales X*, *Chlamydomonadales X*, *Sphaeropleales X*, and *Trebouxiophyceae XX*, all of which are affiliated with the Chlorophyta, and the *Annelida XX* and *Heteroconchia*, which are affiliated with the Metazoa, together constituted the main taxa of the eukaryotic microbial community in the samples. Chlorella species are the best known unicellular green algae. These organisms were frequently used in early studies on photosynthesis [72] and are distributed in both freshwater and marine environments [73,74]. In the present study, *Chlamydomonadales X*, consisting mainly of *Spermatozopsis* and *Chlamydomonas*, were largely distributed in freshwater environments, although *Chlamydomonas* could also be found in seawater environments [75,76,77]. The algae of *Sphaeropleales X* are common planktonic freshwater algae [78], and, in the current study, their members were mainly detected in samples taken from freshwater environments. The eukaryotes of *Trebouxiophyceae XX* detected in this study mainly belonged to *Choricystis*, a photosynthetic endosymbiotic eukaryotic algae commonly found in freshwater sponges and distributed in freshwater environments [79], and were consistent with the distribution range of *Choricystis* in this study. In addition, many taxa of *Annelida XX* and *Heteroconchia* were found to be distributed in brackish and freshwater environments, including *Ficopomatus*, which can tolerate considerable variations in salinity [80], as well as *Lysidice* [81], *Mytilopsis* [82], *Corbicula* [83], *Mercenaria* [84], and some unclassified lineages.

NMDS-based structural analysis of the prokaryotic and eukaryotic communities showed that the community structure differed among sampling sites, with prokaryotic community structure being mainly influenced by rare taxa, while eukaryotic community structure was influenced by a combination of abundant, rare, and common taxa. Generally, microbial communities are usually large in size, but a large number of these taxa are rare and should be very susceptible to ecological drift [26]. Ecological drift refers to the stochastic changes in the relative abundance of different species within a community over time in terms of species characteristics, due to the inherently stochastic processes of birth, death, and reproduction [26,85,86]. Ecological drift, a central concept in community ecology, plays a pivotal role in shaping the structure of microbial communities [87,88,89]. Meanwhile, the distribution of ASVs generated by prokaryotic and eukaryotic communities showed that the unique ASVs were most abundant in pure freshwater environments, such as sampling site 1. Moreover, only a very small fraction of ASVs were shared among all samples. These results suggest that the structure of prokaryotic and eukaryotic communities in the water column was constantly changing as water flowed downstream, thus resulting in different microbial community structures. Based on a larger number of 16S rRNA sequences from different environments around the world, the primary environmental determinant of microbial community composition has been found to be salinity, rather than extremes of temperature, pH, or other physical and chemical factors [90]. In addition to salinity, substantial differences in nutrient availability and types are commonly observed in freshwater and marine environments, and these factors also strongly influence microbial community composition and function [91,92]. For instance, freshwater habitats are predominantly phosphate-limited, while in brackish waters, unique carbon, nitrogen, and sulfur cycling processes dominate, as such activities related to phosphorus acquisition are highest in freshwater environments, along with the dissimilatory nitrate reduction to ammonium in freshwater sediments [92]. In contrast, activities associated with denitrification, sulfur metabolism, and photosynthesis are highest in the brackish zone [92]. Furthermore, eukaryotes in the brackish zone also make a greater contribution to photosynthesis and ammonia oxidation [92].

The results of our neutral community model (NCM) analysis showed that NCM could better predict rare microbial communities compared to abundant and common microbial communities, by estimating the frequency of occurrence of ASVs compared to their relative abundance changes, with higher explanatory rates of community changes. Furthermore, the m values of rare microbial communities were higher than those of common and abundant microbial communities for both prokaryotes and eukaryotes. These findings suggest that the structure formation of rare microbial communities, for both prokaryotes and eukaryotes, is more influenced by stochastic processes and less restricted by species dispersal than that of abundant and common microbial communities. Neutral theory assumes that community structures are regulated by the stochastic processes of birth, death, colonization, extinction, and speciation, rather than species traits [53,93]. Although the relative importance of deterministic and stochastic processes in controlling community structure, succession, and biogeography remains debated [86,88,89,94], it is generally accepted that deterministic and stochastic processes occur simultaneously in local community assemblages [87,95,96,97]. As with species abundance distributions and species-area relationships, some basic ecological patterns in many communities suggest that stochastic processes may play a more important role in generating community patterns than species functional differences [53,98,99,100,101,102].

Furthermore, classical ecological studies suggest that species may differ in their use of multiple limiting resources, and, thus, much of ecology is based on the assumption that species differ in their niches, but it is difficult to infer the underlying ecological processes based on such diversity patterns as species abundance patterns, because different processes or hypotheses can produce very similar, or even the same, diversity patterns [101,102,103]. Thus, even if the observed pattern fits the expectation of the neutral model or null model, we still cannot dismiss the significance of deterministic processes unless we are sure that the deterministic processes cannot yield the same or similar pattern. Even so, the neutral theory is still widely used as a useful null hypothesis or approximation for developing new ecological theories and exploring mechanisms of community assembly [102,103].

## 5. Conclusions

Microorganisms are of fundamental importance for the functioning of riverine, estuarine and marine environments. Although a large number of studies have been conducted, the scope and time span of sampling have been particularly large, and considering the extremely complex physical, chemical, hydrological and anthropogenic disturbances in the river to marine watershed environment, more precise studies of the structural changes of prokaryotic and eukaryotic communities in this system have been relatively rare, especially those for which all samples are collected in a very short period of time. In this study, prokaryotic and eukaryotic communities formed six different community structure groups from the upstream area of the Sanya River to the sea, based on different sampling locations. The structure of prokaryotic communities was mainly influenced by rare microbial taxa, whereas the eukaryotic community structure was influenced by a mixture of abundant, rare, and common microbial taxa. The relative abundance of *Rhodobacteraceae* and *Flavobacteriaceae* increased with distance downstream the Sanya River, while the *Sporichthyaceae* and *Comamonadaceae* were decreased. Some prokaryotic taxa, such as *Phycisphaeraceae* and *Chromobacteriaceae*, were present almost exclusively in pure freshwater environments, while others, including *SAR86 clade*, *Clade I*, *AEGEAN-169 marine group*, and *Actinomarinaceae*, were barely present in pure freshwater environments. The eukaryotic communities were mainly composed of the *Chlorellales X*, *Chlamydomonadales X*, *Sphaeropleales X*, *Trebouxiophyceae XX*, *Annelida XX*, and *Heteroconchia*. Furthermore, the α-diversity of the prokaryotic and eukaryotic communities decreased gradually with distance downstream. The structure formation of rare communities, whether prokaryotic or eukaryotic, was more influenced by stochastic processes and less restricted by species dispersal than that of abundant and common communities. Overall, our study, by virtue of in situ sampling at different sites from upstream to the sea within a short time period, excluded a large number of disturbance factors and also provided new perspectives on microbial ecology in riverine, estuarine, and marine environments.

## Figures and Tables

**Figure 1 microorganisms-11-00536-f001:**
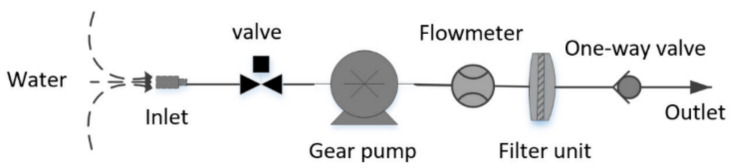
Schematic diagram of the in situ filtration system. During filtration, the gear pump provides the power, the valve opens, and water passes through the filter unit. The flowmeter will record the amount of water sample filtered. The organisms in the water are retained by different criteria depending on the pore size of the membrane.

**Figure 2 microorganisms-11-00536-f002:**
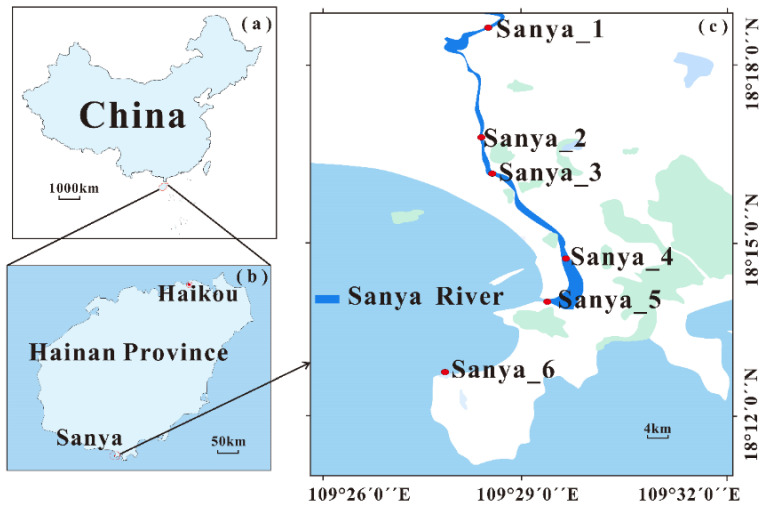
Map of the Sanya River and the sampling sites in this study. The Sanya River is located in Sanya (**c**), Hainan Province (**b**), China (**a**), which is a tropical region. We set up a total of six sampling sites from the upper reaches of the Sanya River to the sea, which we named Sanya 1 to 6, with salinities of 0 ‰, 1 ‰, 6.8 ‰, 22.3 ‰, 33.2 ‰, and 33.5 ‰, respectively.

**Figure 3 microorganisms-11-00536-f003:**
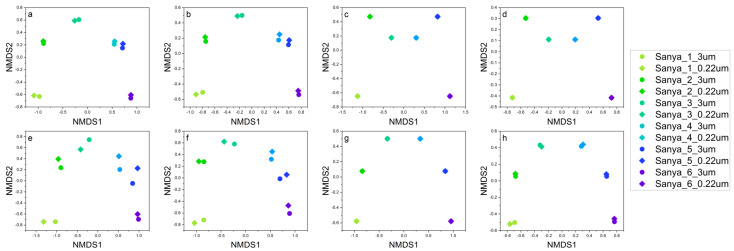
NMDS analysis of the prokaryotic communities separated the samples into six principal groups, which corresponded to the different sampling sites. Furthermore, the structure of the original prokaryotic community (**a**,**b**) was most similar to that of the rare taxa (**e**,**f**) and not similar to that of the microbial community constructed by the abundant (**c**,**d**) and common (**g**,**h**) taxa. The results were based on the ASVs datasets. Plots in (**a**,**c**,**e**,**g**) were calculated based on Bray–Curtis distance, and those in (**b**,**d**,**f**,**g**) were calculated based on Jaccard distance, respectively.

**Figure 4 microorganisms-11-00536-f004:**
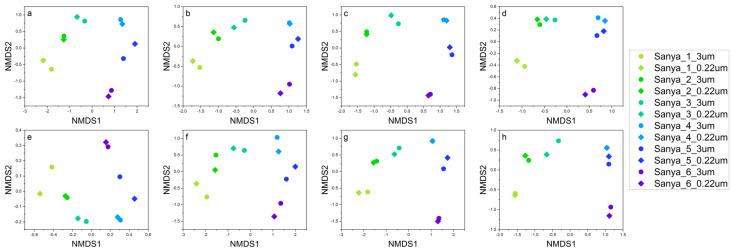
NMDS analysis of the eukaryotic communities separated the samples into six principal groups, which corresponded to the different sampling sites. Furthermore, the structure of the original eukaryotic community (**a**,**b**) was similar to that of the abundant (**c**,**d**), rare (**e**,**f**), and common (**g**,**h**) taxa. The results were based on the ASVs datasets. Plots in (**a**,**c**,**e**,**g**) were calculated based on Bray–Curtis distance, and those in (**b**,**d**,**f**,**g**) plots were calculated based on Jaccard distance, respectively.

**Figure 5 microorganisms-11-00536-f005:**
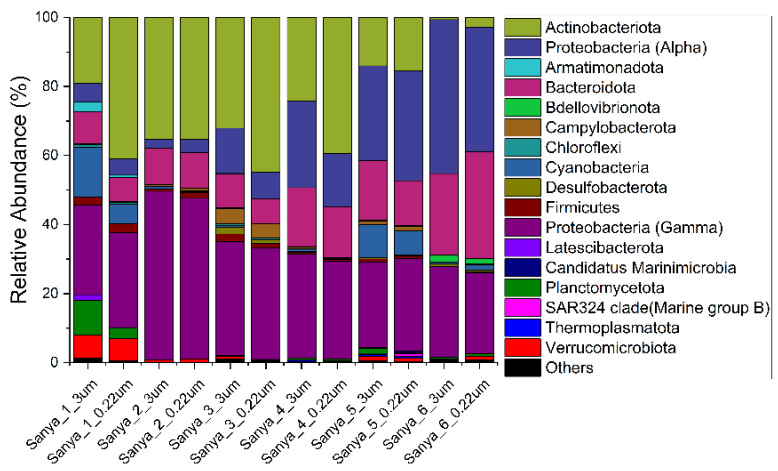
Prokaryotic community members of the 12 in situ filtration samples collected along the Sanya River at the phylum level.

**Figure 6 microorganisms-11-00536-f006:**
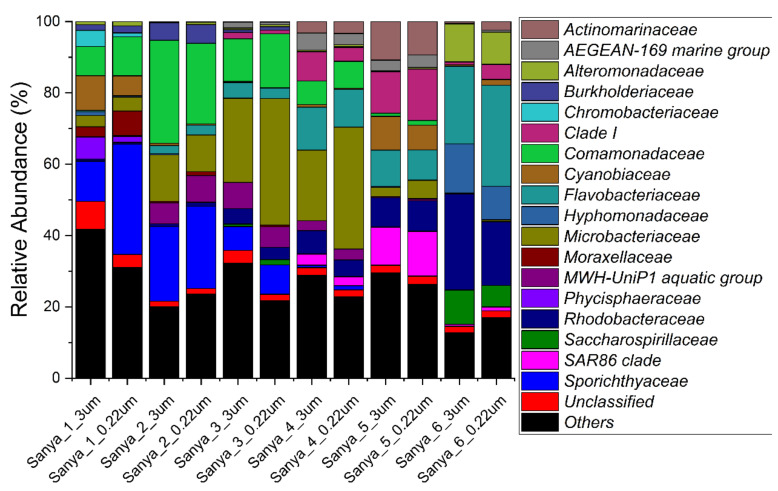
Prokaryotic community members of the 12 in situ filtration samples collected along the Sanya River at the family level.

**Figure 7 microorganisms-11-00536-f007:**
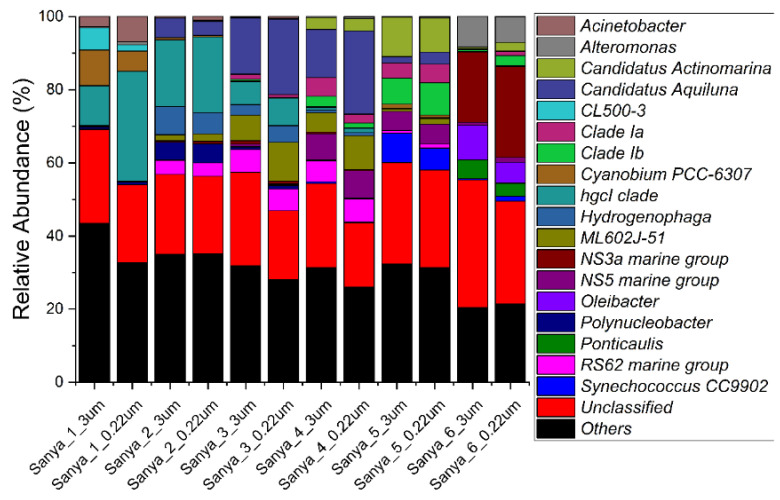
Prokaryotic community members of the 12 in situ filtration samples collected along the Sanya River at the genus level.

**Figure 8 microorganisms-11-00536-f008:**
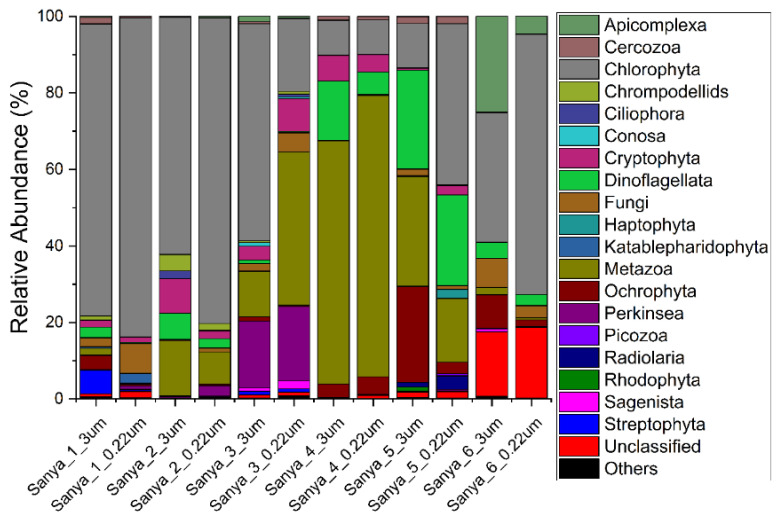
Eukaryotic community members of the 12 in situ filtration samples collected along the Sanya River at the phylum level.

**Figure 9 microorganisms-11-00536-f009:**
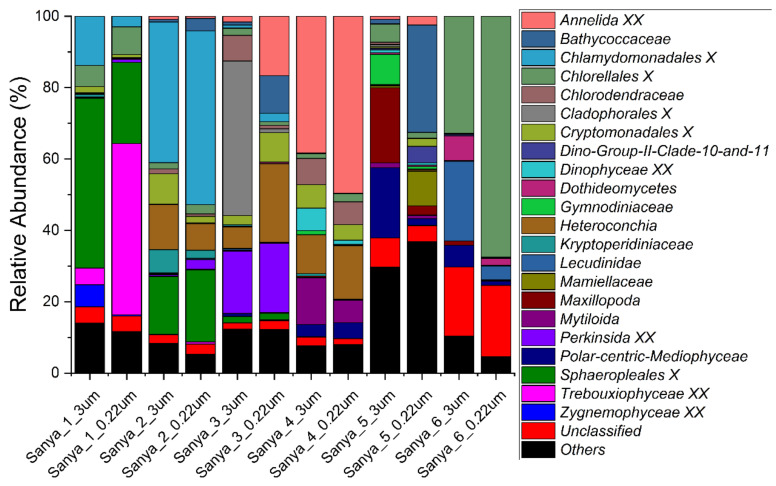
Eukaryotic community members of the 12 in situ filtration samples collected along the Sanya River at the family level.

**Figure 10 microorganisms-11-00536-f010:**
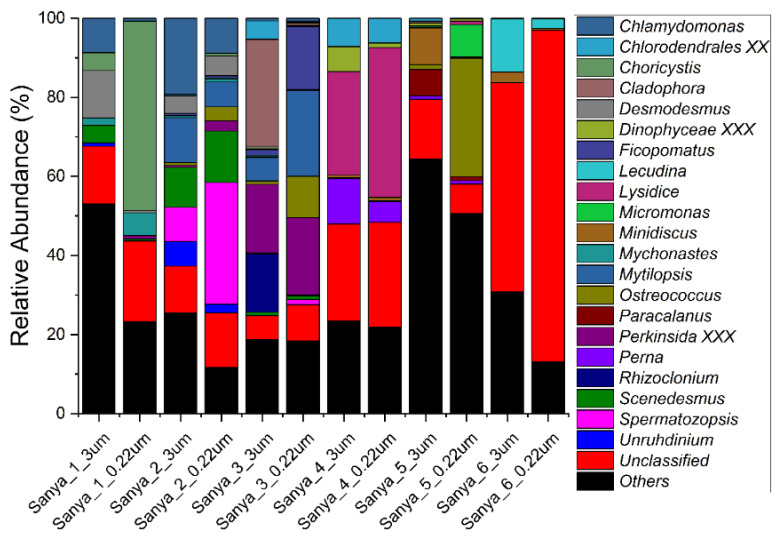
Eukaryotic community members of the 12 in situ filtration samples collected along the Sanya River at the genus level.

**Figure 11 microorganisms-11-00536-f011:**
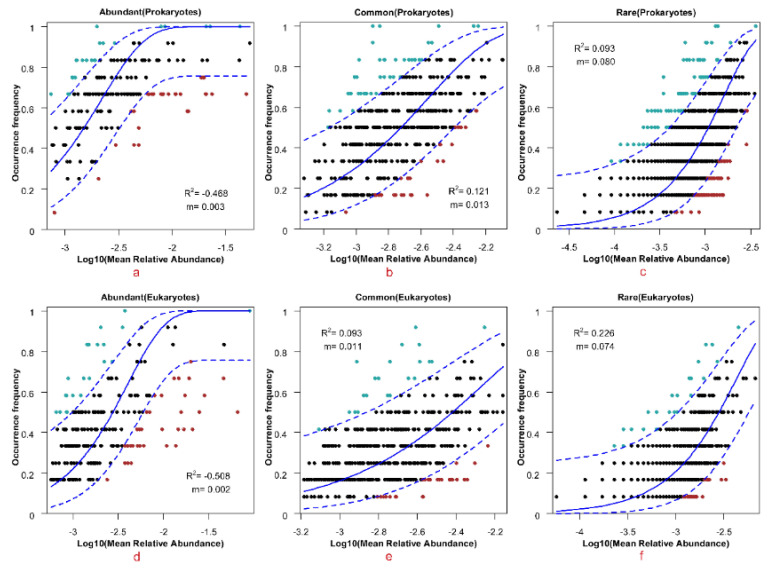
The fit of the neutral community model (NCM) of abundant (**a**,**d**), common (**b**,**e**), and rare (**c**,**f**) community assembly for the prokaryotic and eukaryotic communities. The solid blue line indicates the best fit to the NCM, and the dashed blue line indicates the 95% confidence interval around the model predictions (estimated through 1000 bootstrap). ASVs occurring more or less frequently than the predicted values of the NCM are indicated by different colors. R^2^ indicates the fit to the model, and m indicates the migration rate of the species.

**Table 1 microorganisms-11-00536-t001:** The original (O), abundant (A), rare (R), and common (C) α-diversities of prokaryotic communities from the 12 in situ filtration samples.

Sample Name	Shannon Index				Inv_Simpson Index				Observed Richness				Chao1 Index			
	O	A	R	C	O	A	R	C	O	A	R	C	O	A	R	C
Sanya_1_3 um	5.63	3.89	5.98	5.22	121.31	36.31	315.30	153.60	1008	107	606	295	1136.69	108.67	719.05	313.45
Sanya_1_0.22 um	5.20	3.75	5.72	5.02	68.07	26.98	229.66	123.64	838	103	487	248	966.18	114.25	581.94	265.77
Sanya_2_3 um	4.91	3.87	5.51	4.79	59.99	32.72	172.11	94.12	827	130	451	246	1085.62	141.38	647.87	293.04
Sanya_2_0.22 um	4.69	3.86	5.48	4.75	48.65	31.47	163.53	84.95	760	128	404	228	945.01	136.25	555.54	251.62
Sanya_3_3 um	4.95	3.66	6.01	4.87	40.77	21.05	311.77	88.92	1073	147	640	286	1247.25	154.80	766.91	322.75
Sanya_3_0.22 um	4.66	3.47	5.94	4.98	26.12	14.80	278.13	107.52	1030	155	598	277	1232.58	176.00	736.60	315.33
Sanya_4_3 um	4.63	3.70	5.71	4.64	39.46	24.45	196.82	69.31	920	132	525	263	1177.01	135.21	714.07	335.33
Sanya_4_0.22 um	4.23	3.45	5.53	4.63	21.01	14.73	160.12	68.25	828	141	443	244	1130.58	156.30	727.29	271.03
Sanya_5_3 um	4.88	3.56	5.97	4.79	42.49	20.98	292.49	94.36	969	114	634	221	1176.01	124.11	799.00	248.19
Sanya_5_0.22 um	4.76	3.54	5.74	4.74	40.08	20.62	223.25	91.03	897	125	546	226	1242.24	153.11	774.30	312.25
Sanya_6_3 um	3.99	2.72	5.69	4.52	14.85	8.56	218.60	70.84	719	79	474	166	850.00	88.00	583.00	175.10
Sanya_6_0.22 um	4.16	2.80	5.77	4.69	12.95	7.09	243.78	86.60	758	85	494	179	891.47	90.60	606.50	191.75

**Table 2 microorganisms-11-00536-t002:** The original (O), abundant (A), rare (R), and common (C) α-diversities of eukaryotic communities from the 12 in situ filtration samples.

Sample Name	Shannon Index				Inv_Simpson Index				Observed Richness				Chao1 Index			
	O	A	R	C	O	A	R	C	O	A	R	C	O	A	R	C
Sanya_1_3 um	4.92	3.59	5.41	4.81	58.19	25.42	178.79	103.12	608	84	337	187	689.43	86.63	396.44	206.43
Sanya_1_0.22 um	3.32	2.29	5.05	4.42	5.46	3.60	121.42	62.44	484	78	247	159	583.00	94.50	335.14	165.32
Sanya_2_3 um	4.26	3.48	4.95	4.46	29.80	19.88	99.85	67.56	526	115	232	179	657.21	130.55	290.72	238.13
Sanya_2_0.22 um	3.94	3.26	4.92	4.46	19.22	14.18	95.77	62.23	509	119	226	164	612.89	122.60	305.03	185.94
Sanya_3_3 um	3.28	2.64	4.75	4.05	8.92	6.92	83.51	40.97	439	116	184	139	535.30	134.40	227.56	168.75
Sanya_3_0.22 um	3.41	2.81	4.80	4.05	10.94	8.55	91.37	36.86	464	121	193	150	581.00	123.65	269.61	203.81
Sanya_4_3 um	3.64	2.96	4.49	4.20	11.38	8.29	61.16	51.72	389	95	163	131	541.63	137.17	256.26	148.77
Sanya_4_0.22 um	3.23	2.63	4.56	4.18	6.31	4.95	62.28	48.30	396	90	171	135	560.36	105.17	261.23	184.58
Sanya_5_3 um	4.72	3.70	5.33	4.61	44.91	25.02	156.74	77.22	648	130	330	188	749.97	151.08	384.60	213.09
Sanya_5_0.22 um	4.40	3.32	5.28	4.55	21.59	12.42	159.36	76.25	587	119	300	168	759.22	128.10	408.37	224.10
Sanya_6_3 um	3.68	2.64	5.08	4.26	10.36	6.44	127.78	58.12	453	78	250	125	568.50	93.17	331.36	138.60
Sanya_6_0.22 um	2.06	1.49	4.62	3.90	2.58	2.16	73.16	35.75	329	66	161	102	426.23	106.00	210.34	120.07

## Data Availability

Not applicable.

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
