# Peer review of "Changes of In Situ Prokaryotic and Eukaryotic Communities in the Upper Sanya River to the Sea over a Nine-Hour Period"

_microorganisms, 2023, doi:10.3390/microorganisms11020536_

Round 1

Reviewer 1 Report

Dear authors, the manuscript presents well your research, the background and state of the art and the discussion and comparison with other studies.

Overall, I like the construction of the manuscript and the dimension of each section.

I suggest you to change the keywords to not repeat words from the manuscript title. Also, I suggest you to rewrite the conclusion and present only the main results obtain after this research. The methodological parts should be removed from this section.  

Author Response

Dear Reviewer:

Thank you for your comments on our manuscript entitled "Changes of in situ prokaryotic and eukaryotic communities in the upper Sanya River to the sea over a nine-hour period" (ID: microorganisms-2156204). Your comments are very helpful for revising and improving our paper, as well as the important guiding significance to the next research. We have studied the comments carefully and made corrections which we hope meet with approval. The main corrections are in the new version of manuscript, and the responds to your comments are as follows:

Reviewer 1

Comments and Suggestions for Authors

Dear authors, the manuscript presents well your research, the background and state of the art and the discussion and comparison with other studies.

Overall, I like the construction of the manuscript and the dimension of each section.

I suggest you to change the keywords to not repeat words from the manuscript title. Also, I suggest you to rewrite the conclusion and present only the main results obtain after this research. The methodological parts should be removed from this section.  

Response: Thank you so much. We have made revisions according to your comment. Thanks.

We tried our best to improve the manuscript and made some changes in the manuscript. We appreciate for your warm work earnestly, and hope that the correction will meet with approval.

Once again, thank you very much for your constructive comments and suggestions which would help us to improve the quality of the paper.

Kind regards,

Shijie Bai

Reviewer 2 Report

The authors of the reported study aimed at obtaining realistic information about the microbial community in a particular natural environment (by analyzing the changing patterns of prokaryotic and eukaryotic communities from the river to the sea). In general, the object of this study is within the scope of the journal, however, working with a small number of samples (only two filter samples for each sampling site) is not in agreement with aiming for results of good scientific quality. Please increase the number of samples.

Major comments:

1) The samples were taken only in the winter, in January; that is, the authors did not consider seasonal changes.

2) L. 167: Were the PCR cycling conditions common for 16S rRNA and 18S rRNA gene amplification?

3) Please use the correct name for phyla. Alphaproteobacteria and Gammaproteobacteria are classes; "Candidatus Marinimicrobia" instead of Marinimicrobia …(Figure 5)

4) Increase the quality and readability of figures 3, 4, 11, and 12.

5) The discussion section should be improved. The beginning of this section is a repetition of the introduction section and the materials and methods section. The authors, summarizing the results of the work, give obvious conclusions that the structure of prokaryotic and eukaryotic communities in the water column changes in the direction from the river to the sea, indicating the expected differences, but little affects the ecological role and the reason for the abundance or scarcity of representatives of microbial communities in one or another point of the studied water systems.

6) Please add physical and chemical characteristics of water samples.

The authors should work well on the text; there are both spelling and punctuation errors. There are repetitions in the text.

Author Response

Dear Reviewer:

Thank you for your comments on our manuscript entitled "Changes of in situ prokaryotic and eukaryotic communities in the upper Sanya River to the sea over a nine-hour period" (ID: microorganisms-2156204). Your comments are very helpful for revising and improving our paper, as well as the important guiding significance to the next research. We have studied the comments carefully and made corrections which we hope meet with approval. The main corrections are in the new version of manuscript, and the responds to your comments are as follows:

Reviewer 2

The authors of the reported study aimed at obtaining realistic information about the microbial community in a particular natural environment (by analyzing the changing patterns of prokaryotic and eukaryotic communities from the river to the sea). In general, the object of this study is within the scope of the journal, however, working with a small number of samples (only two filter samples for each sampling site) is not in agreement with aiming for results of good scientific quality. Please increase the number of samples.

Major comments:

  • The samples were taken only in the winter, in January; that is, the authors did not consider seasonal changes.

Response: Thanks for the kindly constructive comments. Yes, studying the changes of river-to-marine prokaryotic and eukaryotic communities in different seasons is a very important and significant work. However, the Sanya River is located in the city of Sanya, which is a tropical region where water temperature does not vary very much. As you mentioned, we used an in situ filtration system to collect water samples along the river in a short period of time in order to obtain realistic information about the microbial community in a given natural environment. Your suggestion is very important and in the future we will develop more in-situ filtration units and then collect enough replicate samples at different sampling sites in different seasons, and we will also try to add different environmental parameter sensors to the in-situ filtration units to collect in-situ environmental parameter information. This current study is a good example and initial attempt to combine engineering R&D and scientific research, and we will continue to improve it in the future. Thank you for your valuable comments, which will provide us with a direction for future research. Thank you.

  • 167: Were the PCR cycling conditions common for 16S rRNA and 18S rRNA gene amplification?

Response: Thank you for pointing this out. After our reconfirmation, the prokaryotic and eukaryotic PCR cycling conditions in this study were the same, only that 27 PCR cycles were performed for prokaryotes and 32 PCR cycles were performed for eukaryotes. We have added some relevant explanation in the section of “2.2. DNA extraction and sequencing”. Thanks.

  • Please use the correct name for phyla. Alphaproteobacteria and Gammaproteobacteria are classes; "CandidatusMarinimicrobia" instead of Marinimicrobia …(Figure 5)

Response: We have made revisions according to your comment. Thank you.

  • Increase the quality and readability of figures 3, 4, 11, and 12.

Response: Thank you for pointing that out. We have readjusted those figures, thanks.

  • The discussion section should be improved. The beginning of this section is a repetition of the introduction section and the materials and methods section. The authors, summarizing the results of the work, give obvious conclusions that the structure of prokaryotic and eukaryotic communities in the water column changes in the direction from the river to the sea, indicating the expected differences, but little affects the ecological role and the reason for the abundance or scarcity of representatives of microbial communities in one or another point of the studied water systems.

Response: Thanks very much for the reviewer’s suggestion and we appreciate. Yes, it is important to understand the ecological role and the reasons for the abundance or scarcity of microbial communities represented in one or another point of the studied water systems. However, attempts to find the reasons for these phenomena need to be combined with information on the function of these microorganisms, and although some software is currently available to predict the function of these microorganisms by using 16S rRNA sequences, such as PICRUSt and FAPROTAX, there are still biases. Therefore, it is necessary to use metagenomics or transcriptomics to reflect the functions of these microorganisms more accurately. We have plans to carry out metagenomic work based on these samples in the future, thank you.

  • Please add physical and chemical characteristics of water samples.

Response: Thank you for your comments. In the future, we will continue to improve our in-situ filtration system by adding various environmental parameters detection sensors to collect the in-situ environmental parameters corresponding to these microorganisms.

The authors should work well on the text; there are both spelling and punctuation errors. There are repetitions in the text.

Response: Thank you for pointing this out. We have sent our manuscript to a specialized language polishing company for revision, and we believe that the writing of the new version will be greatly improved, thank you.

We tried our best to improve the manuscript and made some changes in the manuscript. We appreciate for your warm work earnestly, and hope that the correction will meet with approval.

Once again, thank you very much for your constructive comments and suggestions which would help us to improve the quality of the paper.

Kind regards,

Shijie Bai

Round 2

Reviewer 2 Report

The authors made some changes to the work; unfortunately, the water parameters for complex analysis were not included in the work, and the number of analyzed samples was small. The discussion was not improved. And many responses like "we will do this in the future" spoil the overall picture of responses. Once again, we ask the authors to check the quality of the Figures in their work.

Author Response

Dear Reviewer:

Thank you very much for your suggestions and comments. In fact, after reading your comments, we have actively thought about your suggestions and questions, and it is true that we could not go again to collect the environmental parameters consistent with that time and the replicate samples consistent with that time, which is indeed our shortcoming in this article, but we hope that we can provide some information and reference for better development of river microbial community research by using the idea of combining science and engineering, so that in future We hope that we will be able to design better experiments and conduct large-scale studies in the future. Your suggestions are really important, and we did not fail to answer your questions seriously.

    Regarding the quality of the figures, we have tried various ways to combine these figures into one figure, but they are still not very clear, so we have split them in the new version of the manuscript and unified them as supplementary materials in the manuscript. Thank you so much!

Sincerely,

Shijie Bai
